# Repression of Acetaminophen-Induced Hepatotoxicity in HepG2 Cells by Polyphenolic Compounds from *Lauridia tetragona* (L.f.) R.H. Archer

**DOI:** 10.3390/molecules24112118

**Published:** 2019-06-04

**Authors:** Samuel Odeyemi, John Dewar

**Affiliations:** Department of Life and Consumer Sciences, College of Agriculture and Environmental Sciences, University of South Africa, Johannesburg 1709, South Africa; dewarj@unisa.ac.za

**Keywords:** *Lauridia tetragona*, antioxidant, hepatoprotective, acetaminophen, cell viability

## Abstract

*Lauridia tetragona* (L.f) R.H. Archer is routinely used in traditional medicine; however, its hepatoprotective property is yet to be scientifically proven. To this effect, the hepatoprotective activity of the polyphenolic-rich fractions (PPRFs) was investigated against acetaminophen (APAP) injured HepG2 cells. The ability of the PPRF to scavenge free radicals was tested against 2,2-diphenyl-1-picrylhydrazyl (DPPH), and [2,2′-azino-bis (3-ethylbenzothiazoline-6-sulfonicacid)] (ABTS). The ferric ion reducing power (FRAP) was also evaluated as a cell-free antioxidant assay. The hepatoprotective activity was then investigated by observing the effect of PPRFs against APAP-induced reduction in cell viability of HepG2 cells. The concentrations of alanine aminotransferase (AST), aspartate aminotransferase (ALT) and lactate dehydrogenase (LDH) released into the medium were evaluated while the underlying mechanism was further explored through western blot analysis. Thereafter, the isolated PPRFs were identified using UHPLC-QToF-MS. All six fractions of the PPRFs isolated showed significant antioxidant properties that were evident by the effective scavenging of DPPH, ABTS, and higher FRAP. The results indicated that PPRF pretreatments ameliorated APAP-induced hepatocellular injury by significantly inhibiting the leakage of AST, ALT, and LDH into the medium. The most active fractions for hepatoprotection were PPRF4 and PPRF6 with IC_50_ of 50.243 ± 8.03 and 154.59 ± 1.9 μg/mL, respectively. PPRFs markedly increased activities of liver superoxide dismutase, total antioxidant capacity, and liver glutathione concentration. Both PPRF4 and PPRF6 significantly increased the expression of Nrf2 and translocation. The LC-MS analysis revealed the presence of a wide variety of polyphenolics such as coumarin, ferulic acid, and caffeine among the dominant constituents. In conclusion, this study demonstrates that the isolated PPRFs have potential hepatoprotective activity that may be due to the increased expression of antioxidative genes dependent on Nrf2.

## 1. Introduction

The liver is one of the main sites for the regulation of major bodily functions such as metabolism, storage, detoxification, and secretion of drugs; therefore, any damage to the liver is often associated with distortion of these functions [1]. Hepatic injury is often caused by xenobiotics such as alcohol and chemicals have been associated with redox imbalance and oxidative stress [2]. Oxidative stress occurs when there is an imbalance between the production of reactive oxygen species (ROS) and the antioxidant system in the liver. Recent reports suggest that oxidative stress contributes to the pathogenesis and progression of liver diseases. Hence, it has been suggested that antioxidants can be used for the management of liver injury [3]. Acetaminophen (*N*-acetyl-p-aminophenol, APAP, or paracetamol, PARA) is considered to be nontoxic to the liver at lower doses, but, at higher doses or overdoses, it is highly toxic to the hepatocytes and can cause other cell damage [4]. Conversely, reports over the past few decades implied that there is a connection between the ingestion of therapeutic doses of acetaminophen and liver injury. Likewise, an increase in liver injury markers has been associated with therapeutic doses (≤4 gd^−1^) of acetaminophen [5,6,7].

Acetaminophen is rapidly metabolized in the liver and mostly converted to pharmacologically inactive glucuronide (40–67%) and sulfate (20–46%) conjugates with a small percentage being oxidized to a reactive metabolite *N*-acetyl-p-benzoquinoneimine (NAPQI) (5–10%) [8]. NAPQI is responsible for the hepatotoxicity of acetaminophen. It is inactivated and detoxified by binding to the sulfhydryl group of glutathione (GSH) to form APAP-GSH before it is ultimately excreted as cysteine and mercapturic acid conjugates (APAP-cys) in the urine [9]. Above the therapeutic dosage, excess NAPQI is produced, depleting the GSH stores and causing saturation of the glucuronidation pathway, thereby, leading to the formation of protein adducts through binding to cysteine groups in cellular proteins, primarily mitochondrial proteins and ion channels, thereby, leading to the loss of energy production, ion misbalance, and cell death [4].

There have been concerns regarding the safety of medicinal plants because many of these plants have not been properly investigated scientifically to determine their safe dosages or mutagenicity resulting from their long term use [10]. On this note, one of the medicinal plants that is frequently used by the traditional healers in South Africa for the treatment of diabetes is *Lauridia tetragona* (L.f.) R.H. Archer (Celastraceae). The fruit of this plant is also eaten by children and by birds, while the foliage is sometimes eaten by sheep [11,12]. *L. tetragona* has been reported nontoxic at 5000 mg/kg in Wister rats [13]; however, there is no single report at the time of this study that demonstrates the hepatoprotective potential of *L. tetragona* against acetaminophen-induced liver toxicity. 

In this study, we aimed to evaluate the hepatoprotective activity of the polyphenolic-rich fraction of *L. tetragona* on APAP-induced liver toxicity in HepG2 cells. Furthermore, in order to elucidate the possible mechanism of action, the active fraction was screened for antioxidant activity using FRAP, 2,2-diphenyl-1-picrylhydrazyl (DPPH), and {2,2′-azino-bis(3-ethylbenzothiazoline-6-sulfonicacid)} (ABTS) radical scavenging assays and enzymatic markers for liver damage.

## 2. Results

### 2.1. Polyphenolic Contents

The fractionation of the methanolic extract of the aerial part of *L. tetragona* gave six fractions labeled polyphenolic-rich fraction (PPRF)1–6. All six fractions tested for the phenolic contents showed the presence of polyphenols (Table 1). However, it was found that PPRF3 contained the maximum amount of phenolic compounds with 26.47 ± 1.00 µg QE/ mg of sample. 

### 2.2. Antioxidant Assay

The in vitro antioxidant capacities were estimated with DPPH, ABTS, and FRAP (Table 2). Firstly, with respect to the ability of all the fractions to scavenge DPPH radical, PPRF5 showed better scavenging activities with lower IC_50_ (99.59 ± 13.92 µmol AAE/mL) followed by PPRF3 with 124.153 ± 12.39 µmol AAE/mL. Furthermore, the ABTS scavenging activities of the PPRF3 was better with 251.51 ± 12.39 µmol AAE/mL followed by PPRF5 with 258.11 ± 8.32 µmol AAE/mL. Consequently, PPRF3 also showed better FRAP with 1262.13 ± 32.94 µg AAE/mL compared to the other fractions.

### 2.3. Cell Viability

The cell viability of HepG2 was checked after 24 h treatment with different doses of the PPRFs and APAP (Figure 1 and Figure 2). This was carried out to check for the cytotoxic properties of the PPRFs and APAP, also to confirm the correct dosage required for the hepatoprotective assay. There was a dose-dependent reduction in the percentage viability of the cells treated with PPRFs, except PPRF1 and PPRF2, both of which showed an increase in cell viability. Although there was no significant difference between the increase observed in PPRF1 (250 µg/mL), PPRF2 (62.5 µg/mL), and the reduction in cell viability in PPRF5 at 250 µg/mL. This result showed that the reduction of MTT was insignificant and not cytotoxic at the tested concentrations. Therefore, the non-cytotoxic concentrations were selected for the hepatoprotective assay. As shown in Figure 2, there was a significant reduction in the viability of the cells treated with APAP compared to the control (untreated). 

### 2.4. Inhibition of Acetaminophen-Induced Hepatotoxicity

The possible hepatoprotective effects of the PPRFs against acetaminophen-induced cytotoxicity were determined by pre-incubating the cells with or without the PPRFs. APAP at a dose of 100 mM decreased cell viability to 10% of the control cells. Significant dose-dependent protection toward cell toxicity resulting from APAP exposure was observed in the treated cells with the PPRFs (Figure 3). PPRF4 showed the highest percentage of protection with 77% of the control at 250 μg/mL compared to 62% shown by PPRF6 at 250 μg/mL, whereas PPRF2 and PPRF5 showed the least protection at 250 μg/mL, with 18% and 19%, respectively. 

PPRF4 showed the least half maximal inhibitory concentration of 50.243 ± 8.03 μg/mL followed by PPRF6 with 154.59 ± 1.9 μg/mL. The IC_50_ of all tested samples follows the order PPRF4 > PPRF6 > PPRF1 > PPRF3 > PPRF5 > PPRF2 (Table 3).

### 2.5. Determination of Enzymatic Activities

The treatment of the cells with APAP significantly decreased cell viability and significantly increased the levels of AST, ALT, and LDH in the media compared to the control (Table 4). The cells, when treated with different concentrations (125, 250, and 500 µg/mL) of the PPRFs, showed significant restoration of the LDH leakage towards the untreated control compared to the APAP-treated cells and in a dose-dependent manner. PPRF4 and PPRF6 significantly inhibited (*p* < 0.05) the increase of AST levels in the media following the treatment with APAP at all the treatment doses. However, only PPRF4 significantly inhibited (*p* < 0.05) the increase of ALT levels in the media after APAP treatment at all the doses. PPRF3 significantly inhibited (*p* < 0.05) the increase of AST and ALT levels in the media at only 500 and 250 µg/mL, whereas PPRF2 significantly inhibited (*p* < 0.05) increase of AST and ALT levels in the media only at the highest dose of 500 µg/mL. This indicated that PPRF4 at 500 µg/mL potentially showed the highest percentage inhibitory activity against the increase in the media levels of AST and ALT that was induced by the APAP treatment.

### 2.6. Superoxide Dismutase (SOD) Activity

The treatment of the HepG2 cells with APAP significantly (*p* < 0.05) decreased the SOD concentration compared to the untreated. All the PPRF-treated cells showed a significant increase in the SOD levels, except PPRF3 and PPRF5, both of which were not significantly different compared to the APAP-treated cells (Figure 4). 

### 2.7. Total Antioxidant Capacity (TAOxC)

The treatment of the HepG2 cells with APAP significantly decreased the level of the antioxidant capacity compared to the untreated as shown in Figure 5. All the PPRFs significantly increased the antioxidant capacity; however, only PPRF4 and PPRF6 significantly (*p* < 0.05) increased the antioxidant capacity to a level higher than the untreated cells. 

### 2.8. Reduced Glutathione (GSH) Content

The intracellular GSH was measured to determine the oxidative capacity. There was a significant decrease in the levels of GSH of the cells treated with APAP compared to the untreated cells (Figure 6). The pretreatment of the cells with PPRF1,2,4, and 6 showed a significant increase in the GSH content, whereas no significant increase was observed in PPRF3 and PPRF5 compared to the APAP-treated control cells. 

### 2.9. HO-1 and Nrf2 were Involved in the Protective Activity of PPRFs against APAP-Induced Hepatotoxicity

To investigate whether the protective effects of the PPRFs are associated with the induction of Nrf2 and the HO-1 signaling pathway, the expression of HO-1 and nuclear Nrf2 was measured in APAP injured HepG2 cells pretreated with or without PPRFs. Compared with the control, APAP treatment decreased the expression of HO-1 and Nrf2 but was reversed by the PPRF treatments (Figure 7A). However, only PPRF4 and PPRF6 increased significantly (*p* < 0.05) the expression of HO-1 and Nrf2 (Figure 7B). Furthermore, it was observed that the levels of nuclear Nrf2 were higher than the cytoplasmic Nrf2 (Figure 7C).

### 2.10. Compounds Identified in the PPRFs

The UHPLC-QToF-MS analysis using Bruker software programs such as Data Analysis v4.3 and Profile Analysis v2.0. The identified compounds by the LC-MS detected within the *m*/*z* range of 100–1700 was analyzed using PCA to identify any outliers and assess any groupings or trends. The principal component 1 (PC1) showed that PPRF4 and PPRF6 produced a different pattern of compounds that were significantly different from those of other PPRFs. Principal component 2 (PC2), though, further showed that PPRF2, PPRF4, PPRF5, and PPRF6 were metabolically different from the other PPRFs. The extracted ion chromatograms of all compounds identified are presented in Figure 8 while the total ion chromatograms for the PPRFs are presented in Appendix A. PCA loadings through the compound crawler showed that all the PPRFs contained Diethyl phthalate (Table 5). PPRF2,4,5, and 6 contained ferulic acid, herniarin, varenicline, and 3-Isopropylcatechol, while only PPRF 4 and 6 contained coumarin, 3-tert-butyl-5-methylcatechol, Sterculic acid, and caffeine. Structural elucidation of the identified compounds was done using online libraries including KEGG, PubChem, and ChemSpider. 

## 3. Discussion

Liver diseases remain one of the most serious health problems because there are still no satisfactory liver protective drugs in allopathic medical practice to treat serious liver disorders; however, medicinal plants have been reported to play a major role in the management of various liver disorders by accelerating the natural healing processes of the liver [14]. Although the phytochemicals of *L. tetragona* has not been well documented, its traditional usage suggests medicinal potential. *L. tetragona* is considered beneficial against various diseases namely cancer, diabetes, and liver disease. Other useful properties of *L. tetragona* include anti-tumor, anti-inflammatory, anti-bacterial, immunomodulatory, and analgesic activities [13]. In this study, we evaluated the protective effect of *L. tetragona* against APAP-induced cytotoxicity in HepG2 cells as a strategy to monitor the hepatoprotective activity of six fractions of the plant without high-end testing. It was assumed that HepG2 cells exposed to APAP will lose cell viability and release liver enzymes into the culture medium [15,16]. Incubating HepG2 cells with 10 µM APAP for 24 h caused a significant loss in the cell viability. Pretreatment with the extracts resulted in a dose-dependent increase in cell viability at concentrations ranging from 62.5 to 250 μg/mL suggesting that the samples were not cytotoxic according to previous definitions of cytotoxicity [17,18]. The IC_50_ of PPRF4 of *L. tetragona* in this study was found to protect against liver injury, which is comparable to silymarin with IC_50_ of 25.36 μg/mL (data not shown). The release of ALT, AST, and LDH into the media is an indication of cellular injury; therefore, the reduction in the level of these enzymes in the media suggests possible repair in the injury caused by the APAP treatment. All the fractions showed good antioxidant potentials that could be related to their phenolic contents. Likewise, the hepatoprotective activity of some plants has been directly linked to their antioxidant properties, which include augmenting the GSH redox cycle via increasing intracellular GSH content and GSH/GSSG ratio [18,19,20,21,22,23,24]. Furthermore, the most active fractions against cell injuries are PPRF4, 6, 1, and 3 (in that order), which correlates with their antioxidant activities. Hence, it can be suggested that the mechanism of hepatoprotection of *L. tetragona* reported in this study could be due to the antioxidant activity and/or polyphenolic contents. In the present study, we found that PPRFs induced the expression of HO-1 and Nrf2 protein levels in APAP injured HepG2 cells. Previous reports suggest that APAP-mediated liver injury is exacerbated in Nrf2-deficient animals due to reduced expressions of antioxidant genes. In addition, constant Nrf2 activation increased the basal hepatic GSH levels and accelerated their recovery after APAP treatment. Recent data also support the alleviation of APAP-induced liver injury through the activation of the Nrf2 antioxidant pathway. The findings in this study suggest that PPRF4 and 6 increased the HO-1 expression and Nrf2 expression and nuclear translocation led to compensatory up-regulation of the Nrf2-mediated expression of antioxidant genes for the synthesis of antioxidant enzymes, which may be responsible for the direct prevention of oxidative stress caused by the APAP treatment. The LCMS data also suggest that the most active fractions contain coumarin, caffeine, and ferulic acid. Coumarin, ferulic acid, and caffeine have been reported to exert hepatoprotective effects [25,26,27,28]. The mechanisms underlying the protective effects of coumarins are through modulating the cellular antioxidant pathway either by preventing ROS generation or the increase of the antioxidant enzyme activity [26]. Ferulic acid exerts hepatoprotection against alcohol-induced liver injury through its antioxidant and anti-inflammatory properties and its ability to regulate the NOX4/ROS-MAPK signaling pathway [28]. Furthermore, caffeine administration has been associated with the increased activity of superoxide dismutase and catalase in the liver and the increased expression of nuclear factor E2-related factor 2 (Nrf2), the prototypical transcription factor involved in the induction of antioxidant enzymes [25]. Hence, it is evident that the hepatoprotection observed in this study is similar to the previously mentioned reports. Therefore, the hepatoprotective activities may be due to the coumarin, ferulic acid, and caffeine contents or the synergistic effects of these compounds. 

It is well established that APAP is metabolized in the hepatocytes via the glutathione (GSH) peroxidase converting APAP to inactive glucuronide [1,29]. The depletion of GSH levels and eventual loss of energy increase toxic metabolites that can attack the membrane phospholipids, proteins, and nucleic acids. Thus, antioxidants that can inhibit the production or termination of the activities of free radicals are important in terms of liver protection. Likewise, any compound that is capable of stimulating the synthesis of antioxidant enzymes is beneficial against chemical-induced injuries by stabilizing the antioxidant systems in the cell. This study clearly demonstrates that *L. tetragona* possesses a significant protective effect against APAP-induced cytotoxicity by the activation of antioxidant enzyme synthesis through the Nrf2 pathway. In conclusion, the results of the present investigation infer that the polyphenolic-rich fractions of *L. tetragona* possess potent antioxidant and hepatoprotective properties, the former probably being responsible for the latter. Hence, these polyphenolic-rich fractions can be beneficial for the treatment of liver damage that may arise from exposure to xenobiotics or chemicals.

## 4. Materials and Methods 

### 4.1. Chemicals

3-(4,5-dimethylthiazol-2-yl)-2,5-diphenyltetrazolium bro-mide (MTT), 2,2-diphenyl-1-picrylhydrazyl, 2,2′-azino-bis (3-ethylbenzthiazoline-6-sulfonic acid), ascorbic acid, DMEM, FBS, and TPTZ were purchased from Sigma (Johannesburg, South Africa). Thin-layer chromatography (TLC) plates were products of Inqaba (Johannesburg, South Africa). All other reagents were of quality grade and were purchased from Sigma (Johannesburg, South Africa) and Inqaba (Johannesburg, South Africa).

### 4.2. Plant Material

*Lauridia tetragonia* was collected and authenticated at the South African Biodiversity Institute (SANBI), Pretoria, South Africa. The leaves of the plant were pulverized and extracted in methanol for 24 h on a mechanical shaker. This was then filtered using Whatman No.1 filter paper and the extract evaporated under reduced pressure at 40 °C.

#### 4.2.1. Fractionation 

The methanol extract was loaded into a glass column containing silica gel 60 (0.063–0.200 nm) that was previously washed with methanol. This was then eluted using different solvent systems (Hexane: Ethylacetate: Methanol: water) with increasing polarity, affording a total 36 fractions before the same fractions were combined to give a final 18 fractions and were dried using Genevac EZ-2 series evaporator. The fractions were then chromatographed by thin-layer chromatography (TLC) with EtOAc/AcOH/H2O (10:2:3). These were observed under UV light at 254 and 365 nm and sprayed with DPPH solution (20 g/L) for UV enhancement of phenolic compounds. Six final active fractions (labeled PPRF1–PPRF6) were obtained. The polyphenolic contents were determined and then analyzed by UHPLC-QToF-MS. 

#### 4.2.2. Determination of Total Phenolic Content

The total phenolics in all the fractions were determined with the Folin–Ciocalteau reagent using the modified methods of Armentano et al. [30] and Odeyemi et al. [31]. Briefly, 50 μL of freshly prepared Folin–Ciocalteau’s reagent (1N) and 50 μL of Na_2_CO_3_ (20% *w*/*v*) were added to 7.5 μL of each fraction (250 μg/mL) that was diluted with 42.5 μL of distilled water and incubated at 40 °C for 20 min. Different concentrations (2–10 µg/mL) of quercetin in distilled water was used for the preparation of the calibration curve. All the absorbance was measured at 725 nm. The total phenolic content was expressed as micrograms of quercetin equivalents per milligram of dry weight (μg QE/mg) of extract.

### 4.3. Antioxidant Assay

#### 4.3.1. 2,2-diphenyl-1-picrylhydrazyl (DPPH) Radical Scavenging Assay

The DPPH radical-scavenging activity was determined using the proposed method by Odeyemi et al. [32]. Briefly, 100 µl of different concentrations of the samples or standard was reacted with a freshly prepared solution of 0.135 mM DPPH radical in methanol. The resulting solution was then vortexed and the decrease in absorbance was measured at 517 nm after 10 min. The percentage inhibition was calculated and results were expressed in µM ascorbic acid equivalent/ml. All determinations were performed in triplicate.

#### 4.3.2. 2,2′-Azino-bis (3-ethylbenzthiazoline-6-sulfonic acid) (ABTS) Radical Scavenging Assay

The 2,2′-azino-bis (3-ethylbenzthiazoline-6-sulfonic acid) radical scavenging activity was determined as previously described by Thaipong et al. [33] with little modifications. Briefly, the reaction stock solutions including 7 mM ABTS and 2.4 mM potassium persulfate solutions that have reacted for 12 h at room temperature in the dark was diluted by mixing 60 ml of methanol with 1 mL of the reaction stock and the absorbance adjusted to 0.708 ± 0.001 units at 734 nm. The samples (1 mL) were then reacted with 1 mL of the ABTS radical solution and the absorbance measured using the spectrophotometer at 734 nm after 10 min. The results were expressed in µM ascorbic acid equivalent per ml of the sample. All determinations were performed in triplicate.

#### 4.3.3. Ferric Reducing/Antioxidant Power (FRAP) Assay 

The antioxidant potential was determined according to Odeyemi at al [34]. The assay was based on the reducing power of an antioxidant to reduce the ferric ion (Fe^3+^) to the ferrous ion (Fe^2+^). Fe^2+^ then forms a blue complex (Fe^2+^/TPTZ), which increases the absorption at 593 nm. Briefly, the FRAP reagent was freshly prepared by mixing acetate buffer (300 mM, pH 3.6), TPTZ (10 mM), and FeCl_3_.7H_2_O (20 mM) at 10:1:1 (*v*/*v*/*v*). This was followed by the addition of 100 µl of the FRAP reagent to different concentrations of the samples. The absorbance was taken at 593 nm after 10 min of incubation. All determinations were performed in triplicates and the IC_50_ values were calculated using Finney software, the values were expressed in µg/mL. The results were expressed in µM ascorbic acid equivalent per gram.

### 4.4. Cell Culture

The human hepatoma cell line (HepG2) was a donation from Professor Monde Ntwasa (University of South Africa). HepG2 cells were cultured in Dulbecco’s modified Eagle’s medium (DMEM) supplemented with 10% fetal bovine serum in a humidified incubator at 37 °C with 5% CO_2_. The cells were allowed to reach 80–90% confluence before they were harvested and seeded at a concentration of 5 × 10^6^ in 3 mL culture media into sterile culture dishes, with a diameter of 3.4 cm. Cells were allowed to attach overnight.

#### Cell Viability Assay

HepG2 cells (5 × 10^6^ cells) were cultivated in a 96-well culture plate. After a 24-h incubation at 37 °C, various amounts of the samples were added to the confluent cell monolayer and incubated for another 24 h. Cell viability was monitored by the MTT colorimetric assay as previously described with modification [11]. Cell viability (%) was calculated by the equation: (OD of compound-treated cells/OD of solvent-treated cells) × 100.

### 4.5. Inhibition of Acetaminophen-Induced Hepatotoxicity

The hepatoprotective activity of the samples prepared in DMSO was evaluated against acetaminophen-induced cytotoxicity in HepG2 cells. The cells were maintained with various concentrations of the samples diluted in the cell culture medium such that the final concentration of the DMSO in the culture media did not exceed 1%. Initial cytotoxicity was carried out by incubating the various samples with HepG2 for 24 h prior to the determination of the hepatoprotective assay to determine the concentration range of the samples. The hepatoprotective assay was thereafter determined as previously described by Lee et al. [35] and Kinjo et al. [36]. Briefly, HepG2 cells were seeded into 96-well plates at a concentration of 5 × 10^6^ cells/well and pre-incubated with the samples for 2 h. The culture media were thereafter replaced with another media containing acetaminophen (100 μM) and was incubated for 3 h before it was washed with PBS. The MTT reagent (0.25 mg/ml) was then added to the cells and incubated for 1 h. The excess MTT reagent was discarded and the formazan crystals formed were dissolved using 100 μL of DMSO. The surviving cells were measured using spectrophotometer at 540 nm. The cell viability of the treatments was calculated in relation to the untreated and the results expressed as the percentage of protection using the following equation:

%Protection = [(Mean sample value − Mean of Acetaminophen treated)/(Mean of cell control − Mean of Acetaminophen treated)] × 100.

### 4.6. Determination of Enzymatic Activities/ Biochemical Parameters of Liver Injury

The liver enzymes in the treatment media were determined to investigate the cytotoxic damage. The activities of aspartate aminotransferase (AST), alanine aminotransferase (ALT), and LDH were assayed using commercially available diagnostic kits from Sigma Aldrich, South Africa with catalog numbers MAK055, MAK052, and TOX7, respectively, according to the manufacturer’s instructions.

### 4.7. Measurement of Superoxide Dismutase (SOD) Activity

SOD activity was measured using a Superoxide Dismutase Assay kit (19160, Sigma-Aldrich, South Africa) according to the manufacturer’s instructions, which uses a colorimetric assay to measure the concentration of water-soluble formazan dye produced from the reduction of WST-1 (Water Soluble Tetrazolium) salt with superoxide anion. The supernatant of the lysed cells was added to each well of a 96-well plate before being incubated at 37 °C for 20 min and the absorbance read at 450 nm. 

### 4.8. Measurement of Antioxidant Capacity (TAOxC)

The total antioxidant capacity (TAOxC,) was measured in lysed HepG2 cells using an Antioxidant Assay kit (CS0790, Sigma-Aldrich, South Africa) according to the manufacturer’s instructions. The kit is based on the ability of antioxidants present in the sample to inhibit the oxidation of 2,2′-azino-bis-3-ethylbenzothiazoline (ABTS) to ABTS+ by ferryl myoglobin. HepG2 cells were seeded at a density of 1×10^6^ cells per flask and treated with 20 mM APAP and various concentrations (62.5–250 μg/mL) of the PPRFs for 24 h. After treatment, the cell pellets were homogenized on ice in 1 mL of cold assay buffer before being centrifuged at 12,000 g for 15 min, at 4 °C. The supernatant of the lysed cells was used to measure TAOxC. Absorbance in the well was measured after 5 min at 405 nm. The scavenging activity was expressed as Trolox equivalents (TEAC) from the Trolox calibration curve.

### 4.9. Measurement of Reduced Glutathione (GSH) Content 

The measurement of the reduced glutathione content was carried out using a glutathione assay kit (CS0260, Sigma-Aldrich, South Africa) according to the manufacturer’s instructions. Briefly, HepG2 cells were seeded as previously described. The cell sample deproteinized with 5% 5-sulfosalicylic acid solution was used to measure GSH content. The absorbance was measured at 405 nm using a microplate reader. The GSH content was determined by comparing the slope of the sample with that of the standard curve.

### 4.10. Protein Extraction

Cells were pre-incubated with or without the PPRFs (500 μg/mL) for 60 min and then incubated with or without APAP for the additional indicated time. After treatment, cellular proteins were extracted, and their concentrations measured using the ReadyPrep™ Protein Extraction Kit (Bio-Rad, Johannesburg, South Africa) according to the manufacturer’s instructions.

### 4.11. Western Blot Analysis

Protein concentrations were measured using a protein assay kit (Bio-Rad, South Africa) according to the manufacturer’s instructions. Extracted proteins were subjected to sodium dodecyl sulfate-polyacrylamide gel electrophoresis (SDS-PAGE) and transferred to polyvinylidenedifluoride (PVDF) membranes. Each membrane was incubated in a TNA buffer (10 mM Tris-Cl, pH-7.6, 100 mM NaCl, and 0.5% Tween 20) containing 5% skim milk for 1 h to block non-specific binding. Each membrane was then incubated with different primary antibodies at 4 °C overnight. After washing membranes for 1 h with the TNA buffer, membranes were incubated with horseradish peroxidase-conjugated anti-mouse or anti-rabbit antibody at room temperature for 30 min. Blots were developed using the GelDoc XRS apparatus (Bio-Rad, South Africa). The protein bands were quantified by the average ratios of integral optic density following normalization to the expression of internal control β-actin or Lamin B, and the results were further normalized to control.

### 4.12. Ultra-High Performance Liquid Chromatography Mass Spectrometry Analysis

An Agilent ultra-high-performance liquid chromatography mass spectrophotometer (Compass QToF Series 1.9, Bruker Instrument: Impact II) system was used for the UHPLC-QTOF-MS analysis. The chromatographic separation was carried out using an Acquity UPLC BEH C18 column 1.7 um, diameter 2.1 × 100 mm (Miscrosep Waters, Johannesburg, South Africa). The mobile phase consisted of formic acid (FA) in water and acetonitrile. The column flow was set at 0.3 mL/min, column oven temp at 35 °C, and draw speed at 3 µl/s with a total injection volume of 2 µL. The parameters for the mass spectrometer (MS) were as follows: capillary voltage 4500 V, drying gas 8 l/m, gas temperature 200 °C, ionization energy 4.0 eV, collision energy 7.0 eV, and cycle time 0.5 s. Data analysis was done using the Bruker Software (Bruker Compass Data Analysis 4.3, Bruker Daltonik GmbH, Bremen, Germany, 2014). Final results were compared by using the online National Institute of Standards and Technology (NIST) LC/MS library.

## Figures and Tables

**Figure 1 molecules-24-02118-f001:**
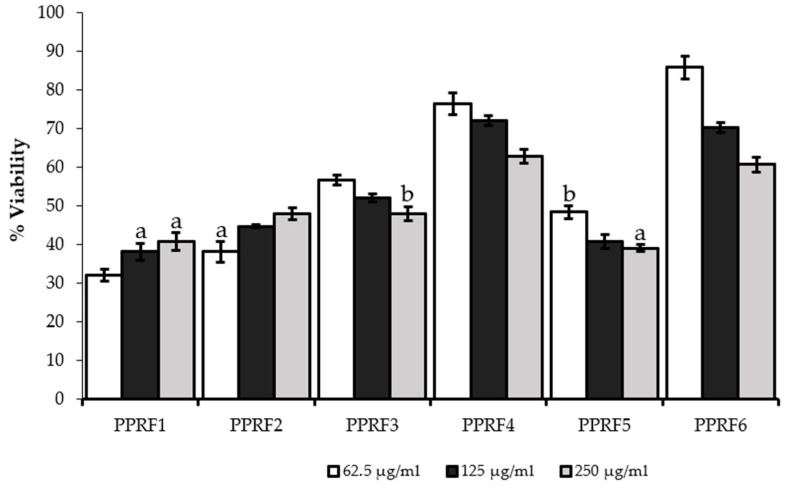
Cytotoxic effect of polyphenolic fractions of *L. tetragona* on HepG2 cells after 24 h treatment. Values are means ± SD percentage of control from three replicates of independent experiments. Bars with the same letters are not significantly different (*p* < 0.05).

**Figure 2 molecules-24-02118-f002:**
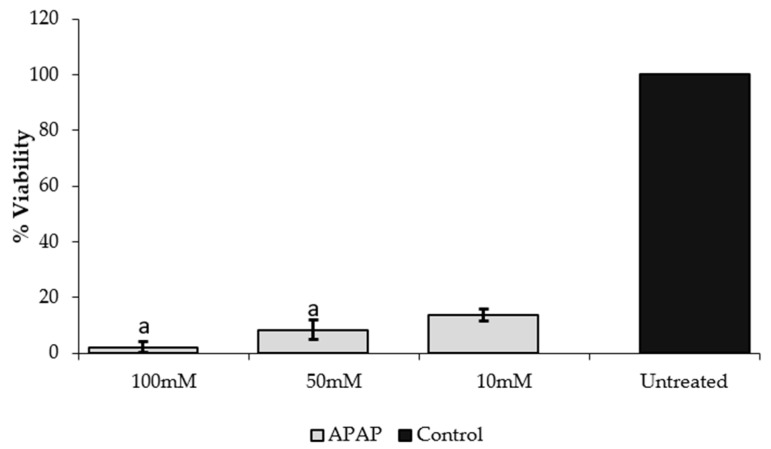
Cytotoxic effect of APAP on HepG2 cells after 24 h treatment. Values are means ± SD percentage of control from three replicates of independent experiments. Bars with the same letters are not significantly different (*p* < 0.05). APAP: Acetaminophen.

**Figure 3 molecules-24-02118-f003:**
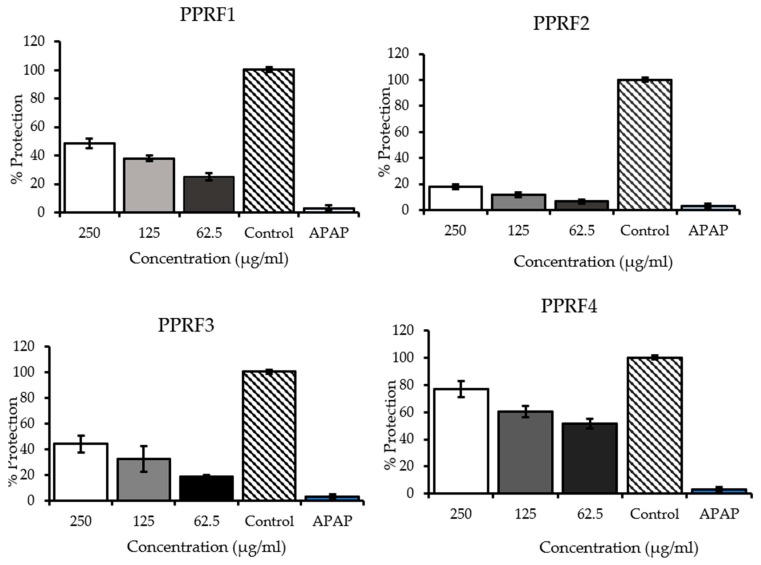
Effect of polyphenolic-rich fractions of *L. tetragona* on acetaminophen (100 mM)-induced cytotoxicity in HepG2 cells. The values are expressed as the mean ± SD percentage of control (*p* < 0.05) from three replicates of independent experiments. APAP: Acetaminophen.

**Figure 4 molecules-24-02118-f004:**
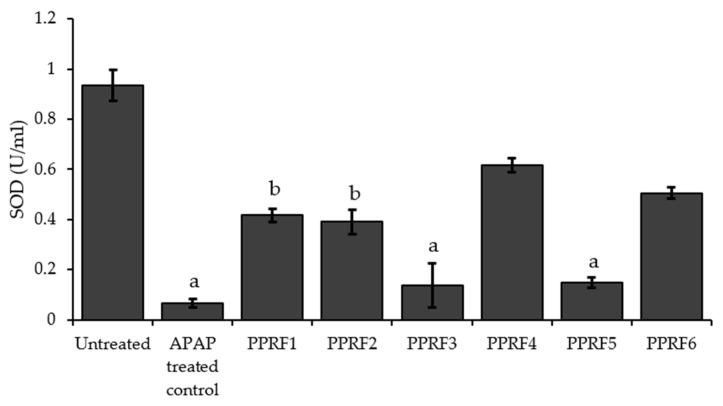
Effect of PPRF treatment on the SOD levels of APAP-treated HepG2 cells. The values are expressed as mean ± SD. Significant differences (*p* < 0.05) from three replicates of independent experiments. Bars with the same letters are not significantly different (*p* < 0.05). APAP: Acetaminophen.

**Figure 5 molecules-24-02118-f005:**
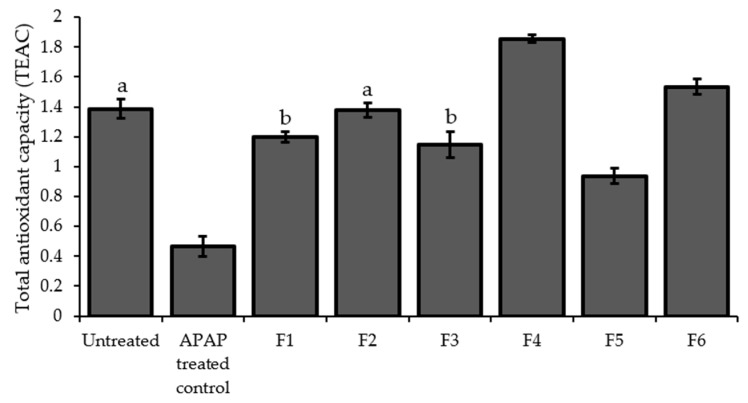
The effect of PPRFs on the total antioxidant capacity of APAP-treated HepG2 cells. The values (mean ± SD) from three replicates of independent experiments and are expressed as Trolox equivalents (TEAC). Bars with the same letters are not significantly different (*p* < 0.05). APAP: Acetaminophen.

**Figure 6 molecules-24-02118-f006:**
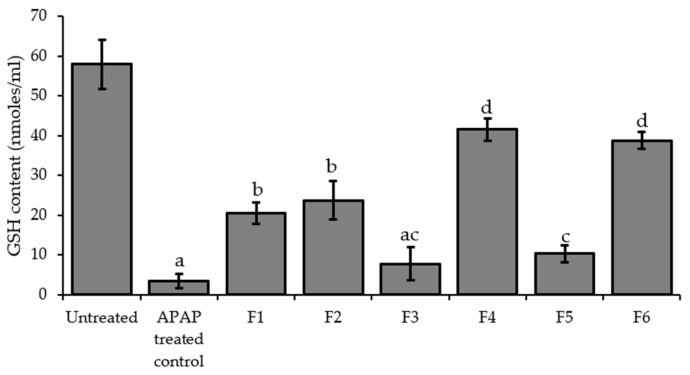
The effect of PPRFs on the GSH content of APAP-treated HepG2 cells. The values (mean ± SD) from three replicates of independent experiments. Bars with the same alphabets are not significantly different (*p* < 0.05). APAP: Acetaminophen.

**Figure 7 molecules-24-02118-f007:**
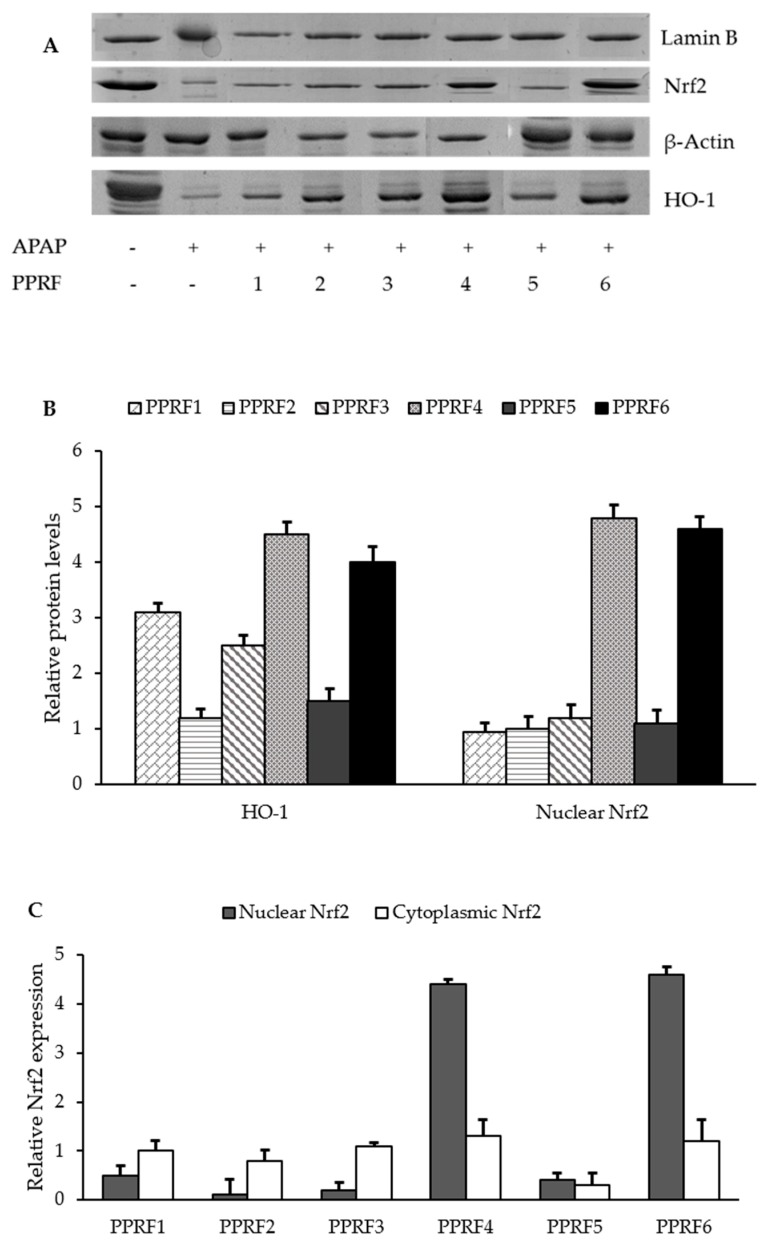
Effects of PPRFs on APAP-induced HO-1, β-Actin and, Nrf2 expression. Cells were treated with APAP (10 Mm) for 36 h and equal amounts of total proteins were subjected to sodium dodecyl sulfate–polyacrylamide gel electrophoresis. (**A**) The expressions of HO-1, β-Actin, Nrf2 protein were detected by Western blotting using corresponding antibodies. Lamin B1 and β-Actin were used as loading controls. (**B**) Relative expression of the proteins (**C**) Treatment with PPRFs for 36 h increased the Nrf2 levels in the nuclear fraction of PPRF4 and PPRF6, whereas the treatment decreased the levels in the cellular fraction. Values are the mean ± SD of three independent experiments. *p* < 0.05 indicates significant differences compared to the control group.

**Figure 8 molecules-24-02118-f008:**
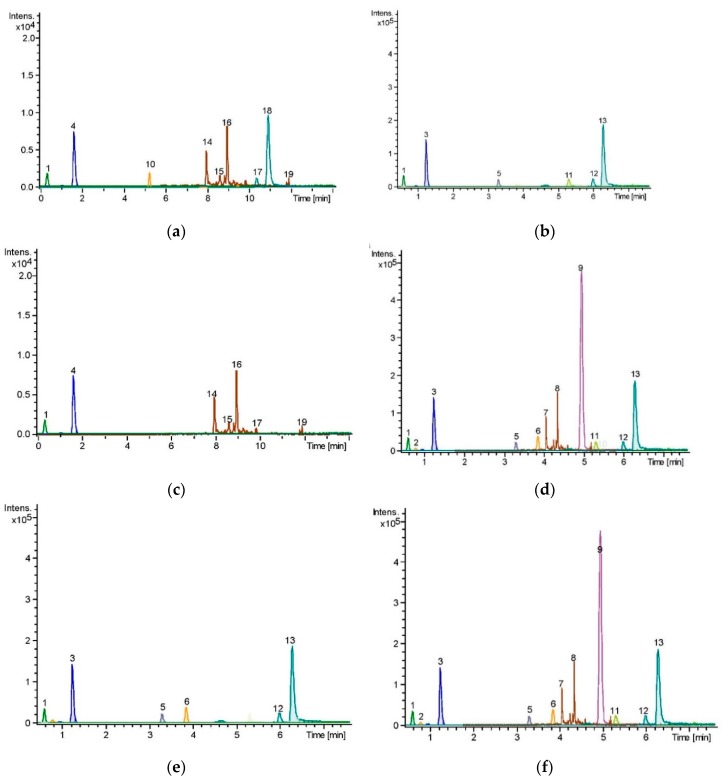
Extracted ion chromatograms (EICs) of the PPRFs by UHPLC-MS. (**a**) PPRF1 (**b**) PPRF2 (**c**) PPRF3 (**d**) PPRF4 (**e**) PPRF5 (**f**) PPRF6.

**Table 1 molecules-24-02118-t001:** Total phenolic content (TPC) of different fractions of *L. tetragona* methanolic extract.

Samples	Phenolic Content
	µg QE/mg of sample
PPRF1	20.51 ± 2.9
PPRF2	16.38 ± 1.08
PPRF3	26.47 ± 1.00
PPRF4	22.13 ± 2.09
PPRF5	15.62 ± 1.93
PPRF6	16.35 ± 2.79

**Table 2 molecules-24-02118-t002:** Antioxidant properties of polyphenolic-rich fractions of *L. tetragona.*

Samples	DPPH IC_50_ (µmol AAE/mL)	ABTS IC_50_ (µmol AAE/mL)	FRAP IC_50_ (µg AAE/mL)
PPRF1	130.459 ± 24.9	310.459 ± 24.9	1761.49 ± 22.4
PPRF2	212.95 ± 26.57	302.95 ± 26.57	1676.44 ± 70.75
PPRF3	124.153 ± 12.39	251.51 ± 12.39	1262.13 ± 32.94
PPRF4	187.23 ± 20.69	296.23 ± 20.69	4423.06 ± 26.06
PPRF5	99.59 ± 13.92	258.11 ± 8.32	2235. 63 ± 21.21
PPRF6	137.77 ± 19.66	351.11 ± 20.31	2220.89 ± 29.62

All the values are represented as mean ± SD; µmol AAE: µmol of Ascorbic Acid Equivalent.

**Table 3 molecules-24-02118-t003:** Hepatoprotective effects of different polyphenolic fractions of *L. tetragona.*

Samples	IC_50_
	μg/mL
PPRF1	251.935 ± 6.7
PPRF2	788.14 ± 4.2
PPRF3	287.893 ± 2.11
PPRF4	50.243 ± 8.03
PPRF5	510.35 ± 3.4
PPRF6	154.59 ± 1.9

**Table 4 molecules-24-02118-t004:** Protective effect of polyphenolic-rich fractions of *L. tetragona* on APAP-Induced hepatotoxicity in HepG2 cells.

Samples	APAP	Dose	AST	ALT	LDH
		(µg/mL)	(mU/mL)	(mU/mL)	
PPRF1	+	500	32.60 ± 3.1 ^#^	29.06 ± 3.6	39.82 ± 4.2 ^#^
	+	250	39.80 ± 9.3	33.39 ± 3.3	44.94 ± 2.5 ^#^
	+	125	44.15± 3.9	35.10 ± 3.5	49.82 ± 2.2 ^#^
PPRF2	+	500	34.24 ± 4.3 ^#^	23.36 ± 3.2 ^#^	34.67 ± 2.5 ^#^
	+	250	41.10 ± 2.1	29.49 ± 2.1	38.62 ± 5.2 ^#^
	+	125	46.35± 4.2	33.49 ± 3.7	40.01 ± 0.2 ^#^
PPRF3	+	500	26.22 ± 3.6 ^#^	20.26 ± 2.2 ^#^	26.01 ± 3.2 ^#^
	+	250	31.40 ± 1.3 ^#^	25.15 ± 1.5 ^#^	27.57 ± 8.1 ^#^
	+	125	36.33± 4.6	29.17 ± 5.2	31.11 ± 4.2 ^#^
PPRF4	+	500	12.98 ± 4.3 ^#^	9.26 ± 4.2 ^#^	27.88 ± 3.1 ^#^
	+	250	19.78 ± 2.1 ^#^	15.48 ± 2.2 ^#^	23.8 ± 4.1 ^#^
	+	125	25.25± 3.2 ^#^	19.96 ± 2.6 ^#^	17.18 ± 1.6 ^#^
PPRF5	+	500	31.44 ± 7.3 ^#^	27.26 ± 2.4	32.19 ± 2.4 ^#^
	+	250	36.51 ± 3.9	31.42 ± 4.4	35.84 ± 4.2 ^#^
	+	125	41.20 ± 9.7	36.33 ± 4.7	39.61 ± 5.3 ^#^
PPRF6	+	500	18.34 ± 4.4 ^#^	25.43 ± 4.9	33.05 ± 0.8 ^#^
	+	250	22.41 ± 4.3 ^#^	31.02 ± 6.2	29.23 ± 3.4 ^#^
	+	125	35.11 ± 3.6 ^#^	38.10 ± 1.5	22.11 ± 2.3 ^#^
Control	-	-	5.20 ± 1.2	3.26 ± 1.7	0.89 ± 2.3
APAP-treated control	+	100 mM	46.50 ± 6.4 *	35.05 ± 5.3 *	65.95 ± 7.3 *

^#^ significantly different (*p* < 0.05) compared with the APAP-treated control; * significantly different (*p* < 0.05) compared with the control. +: present; -: absent. Data are mean ± SD of triplicate and *p* < 0.05 is considered significantly different.

**Table 5 molecules-24-02118-t005:** Retention time (RT), measured mass, and calculated formula by elemental compositions for major compounds identified in the PPRFs.

Compound Number	Major Compounds	Fraction(s)	t_R_ (min)	Formula	Measured Mass (*m*/*z*)	Positive ESI Mode		
						Adduct Ions	Mass Error (ppm)	MS^n^ Fragment Ions
1	Diethyl phthalate	1,2,3,4,5,6	0.2182	C_12_H_14_O_4_	222.24	221.089 [M + H]^+^	0.5	65.039 [C5H4]+H^+^, 93.034 [C6H4O]+H^+^, 121.028 [C7H4O2]+H^+^
2	Homovanillic acid	6	0.8032	C_9_H_10_O_4_	182.17	181.058 [M + H]^+^	1.2	110.036 [C6H4O2+H]+H^+^, 110.036 [C6H4O2+H]+H^+^,
3	Varenicline	2,4,5,6	1.2265	C_13_H_13_N_3_	212. 11	211.267 [M + H]^+^	−0.7	77.039 [C6H4]+H^+^, 94.065 [C6H7N]+H^+^, 119.06 [C7H8N2-H]^+^, 195.092 [C13H11N2]^+^
4	Pedaliin	1,3	1.8622	C_22_H_22_O_12_	479.23	478.112 [M + H]^+^	7.1	83.049 [C5H4O+2H]+H^+^, 83.049 [C5H4O+2H]+H^+^, 111.044 [C6H5O2+H]+H^+^, 111.044 [C6H5O2+H]+H^+^, 139.039 [C7H5O3+H]+H^+^, 139.039 [C7H5O3+H]+H^+^
5	3-Isopropylcatechol	2,4,5,6	3.2675	C9H12O2	153.09	152.084	−0.4	67.05426 [C5H8-H]^+^
6	2-Methoxy-4-vinylphenol	4,5,6	3.829	C_9_H_10_O_2_	151.07	150.177 [M + H]^+^	0.6	103.054 [C8H6]+H^+^, 117.034 [C8H6O-H]^+^, 121.065 [C8H7O+H]+H^+^
7	Caffeine	4,6	4.169	C_8_H_10_N_4_O_2_	195.08	194.19 [M + H]^+^	−0.2	110.035 [C4H6N3O-2H]^+^, 110.035 [C4H6N3O-2H]^+^, 152.058 [C7H7N2O2]+H^+^, 152.058 [C7H7N2O2]+H^+^
8	3-tert-Butyl-5-methylcatechol	4,6	4.2581	C_11_H_16_O_2_	181. 12	180.247 [M + H]^+^	−0.1	93.07 [C7H10-H]^+^, 121.065 [C8H11O-2H]^+^, 149.06 [C9H10O2-H]^+^
9	Coumarin	4,6	5.0014	C_9_H_6_O_2_	147.619	146.037 [M + H]^+^	5.6	77.039 [C6H4]+H^+^, 103.053 [C8H6]+H^+^, 117.034 [C8H6O-H]^+^
10	Gambiriin A2	1	5.193	C_30_H_28_O_12_	581.1654	580.1581 [M + H]^+^	−1.3	503.134 [C28H25O9-2H]^+^, 563.155 [C30H27O11]^+^
11	Gentioflavine	2,4,6	5.2539	C_10_H_11_NO_3_	194.72	193.07393 [M + H]^+^	−1.3	97.028 [C5H5O2]^+^, 112.039 [C5H6NO2]^+^, 127.039 [C6H5O3+H]+H^+^
12	Herniarin	2,4,5,6	6.0091	C_10_H_8_O_3_	177.0546	176.047 [M + H]^+^	−1.1	121.028 [C7H6O2-H]^+^, 149.023 [C8H6O3-H]^+^
13	Ferulic acid	2,4,5,6	6.242	C_10_H_10_O_4_	195.18	194.0594 [M + H]^+^	−0.1	77.0386 [C6H5]^+^, 117.034 [C8H6O-H]^+^, 121.065 [C8H6O+2H]+H^+^, 145.028 [C9H6O2-H]^+^, 149.06 [C9H9O2]^+^, 177.055 [C10H9O3]^+^
14	3-Aminophenol	1,3	8.0365	C_6_H_7_NO	110.13	109.05279 [M + H]^+^	0.8	77.039 [C6H4]+H^+^
15	4-[(2-tert-butoxy-2-oxo-ethyl)-(3-methoxy-3-oxo-propyl)amino]-3-nitro-benzenesulfonic acid	1,3	8.7380	C_23_H_16_O_8_	419.212	418.105 [M + H]^+^	3.5	315.065 [C12H15N2O6S]^+^, 315.065 [C12H15N2O6S]^+^, 330.088 [C13H17N2O6S]+H^+^, 401.101 [C16H22N2O8S-H]^+^
16	4-Piperidone	1,3	9.0382	C_5_H_9_NO	99.13	100.068 [M + H]^+^	10.2	72.081 [C4H9N]+H^+^
17	Coumarin 314	1,3	10.372	C_18_H_19_NO_4_	314.1387	313.131 [M + H]^+^	−1.3	77.0386 [C6H4]+H^+^, 93.07 [C7H6+2H]+H^+^, 103.054 [C8H6]+H^+^, 117.034 [C8H6O-H]^+^, 121.065 [C8H9O]^+^, 145.028 [C9H6O2-H]^+^, 149.06 [C9H9O2]^+^, 177.055 [C10H9O3]^+^
18	Petunidin 3-O-glucoside	1	11.1382	C_22_H_22_O_12_	479.1184	478.111 [M + H]^+^	5.4	85.028 [C4H5O2]^+^, 302.042 [C15H9O7]+H^+^, 317.066 [C16H12O7]+H^+^
19	Venlafaxine	1,3	11.8120	C_17_H_28_NO_2_	278.2115	277.204 [M + H]^+^	−1.7	163.112 [C11H15O]^+^, 219.174 [C15H20O+2H]+H^+^, 222.149 [C13H19NO2]+H^+^,
20	Sterculic acid	4,6	9.94	C19H34O2	295.2632	294.256 [M + H]^+^	−0.2	69.06992 [C5H10-H]^+^, 83.086 [C6H12-H]^+^, 97.101 [C7H14-H]^+^, 111.117 [C8H12+2H]+H^+^, 123.117 [C9H14]+H^+^, 135.117 [C10H16-H]^+^, 149.133 [C11H18-H]^+^, 163.148 [C12H20-H]^+^, 263.237 [C18H30O]+H^+^

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
