# Peer review of "Repression of Acetaminophen-Induced Hepatotoxicity in HepG2 Cells by Polyphenolic Compounds from Lauridia tetragona (L.f.) R.H. Archer"

_molecules, 2019, doi:10.3390/molecules24112118_

Round 1
Reviewer 1 Report
The manuscript suggests some polyphenol fractions from Lauridia tetragonahave hepatoprotective effect. The manuscript was improved very well, and the data are interesting.
Which compound listed in Figure 8 is the active substance?
L211: p<0.05
Author Response
L285 and L289: Indicated that PPRF4 and PPRF6 were the active fractions and both contain Coumarin, caffeine and ferulic acid as the active compounds
L211: p < 0.05 has been changed to p > 0.05
Reviewer 2 Report
Title: Repression of acetaminophen-induced hepatotoxicity in HepG2 cells by polyphenolic compounds from Lauridia tetragona (L.f.) R.H. Archer In the Abstract, the authors have indicated that the isolated PPRFs were identified by using UHPLC-Q-ToF-MS. It was strongly suggested that the analysis results, i.e. total ion chromatograms and characteristics of the chromatograms should be presented in the manuscript. Furthermore, the Figure 8 was unnecessarily shown in the manuscript. In the Discussion, the authors should give some discussion from the cited references for presenting the bioactivities of the identified compounds in the protective function of liver. In the compound identification, the compounds described in Lines 235-239 were very different from the compounds shown in Figure 8. The authors should take note in the differences and give the exact identification. All of the identification data from LC/MS analysis should be appended in a Table to show the mass spectrum. Overall, the authors have completed a reasonable study with some informative data on the liver protective function of Lauridia tetragona. However, some of major concerns described in the above would be suggested and requested for further improvement in the manuscript.

Author Response
Attached please find the responses to your comments. Since all the LC/MS analytical data are too large to be appended to the manuscript, these will be made available upon request.

Reviewer 3 Report
The authors made an apprecciable effort to integrate new data.
The manuscript still requires revision before becoming acceptable for publication.
1. The authors keep having problems with the definition of statistically significant data. See for example the legend to
Fig.1: ".... alphabets are not significantly different (? < 0.05) while bars without any alphabet are significantly different (? < 0.05)."
and Fig. 2: "s. Bars with the same alphabets 108 are not significantly different (? < 0.05) while bars without any alphabet are significantly different (? < 0.05)."
Significantly different or not significantly different data cannot both correspond to p<0.005!
The same problem applies to Fig. 5 and Fig. 6
Fig. 3: How can the reader recognize the data associated to p<0.005 ?
2. Language revision is needed. For example the sentence at page 6, lines 128-129 should read:
"The treatment of the cells with APAP significantly decreased cell viability and significantly increased the levels of AST, ALT and LDH in the media compared to the controls (Table 4)."
In the following sentences of the same paragraph, the use of some articleas would help readability.
Fig.3, legend: the following sentence should be rephrased because not understandable in its present form: "Significant differences (? < 0.05) percentage of control from three replicates of independent experiments."
Author Response
Attached please find the responses to your comments

Round 2
Reviewer 2 Report
Manuscript ID: molecules-435190
Title: Hepatoprotective effect of polyphenolic rich fractions from Lauridia tetragona (L.f.) R.H. Archer on acetaminophen induced liver injury
In this study, the polyphenolic extracts of Lauridia tetragona (1,4-GL) were applied to investigate the prevention effects on APAP-induced hepatotoxicity in HepG2 cell model. The GC-MS-based analysis method was used to investigate the polyphenolic extracts.
In the last reviewed results, the compounds analysis has been questioned due to the inadequacy of GC/MS analysis in the phenolic extracts. The authors should address how to analyze the compounds and give the data in the Table scientifically, including the UV-Vis absorption wavelength, the m/z of negative ESI and the exact measured mass. It was noticed that most of the data in Table 5 was from tentative identified result. If the authors could list the mass fragmentation from MS/MS analysis in the Table, it would be more completely convinced. Moreover, if the works were not familiar with the authors, it was suggested that the authors should ask for help from the expert of LC/MS analysis.
The second question was newly found from the raw material. Which part of the plant was used in the study, fruits or leaves or all parts?
Author Response
Attached is the response to your comments

This manuscript is a resubmission of an earlier submission. The following is a list of the peer review reports and author responses from that submission.
Round 1
Reviewer 1 Report
The manuscript suggests some polyphenol fractions from Lauridia tetragona have hepatoprotective effect. The data are interesting. However, there are some concerns that should be addressed. This article would be reconsidered after revision.
What is the most potent substance in Lauridia tetragona? How about the activity of ferulic acid, carvone, stearic acid, and so on contained in the Lauridia tetragonaextracts?
What is the mode of actin of the hepatoprotective effect of PPRFs? Just only antioxidative activity? If so, why PPRF5 which has higher antioxidant activity is no so effective on hepatoprotection? Are there any other reasons for hepatoprotective effect of PPFRs?
The title should be changed. Because no in vivoexperiment on liver injury was conducted. Just in vitro experiments using liver cell line.
Reviewer 2 Report
Manuscript ID: molecules-435190
Title: Hepatoprotective effect of polyphenolic rich fractions from Lauridia tetragona (L.f.) R.H. Archer on acetaminophen induced liver injury
In this study, the polyphenolic extracts of Lauridia tetragona were applied to investigate the prevention effects on APAP-induced hepatotoxicity in HepG2 cell model. The GC-MS-based analysis method was used to investigate the polyphenolic extracts. Results indicated that different fractions showed significantly different antioxidant activities and cell protective activities. Overall, the authors have completed part of the study to present some of the informative data on the prevention effects of extracts. However, the major concerns in the compounds analysis and the key mechanisms of protective effect would be requested for further improvement in the manuscript.
1. P. 293. The GC-MS analysis
In the GC/MS analysis procedures, the gradient procedures (P. 300-) shown in the section should be the HPLC analysis conditions not the GC/MS analysis conditions.
2. Table 4.
The TIC of GC/MS shown in the Table 4 should be separated and presented with a figure to indicate the peak number for explaining the compound name corresponding to the Table 4.
3. The protective mechanisms of extracts on APAP-induced toxicity should be conducted further by western blot analysis to explain the possible reason.
4. Finally and more importantly, the analysis of extracts should be conducted by using LC/MS, especially in the solvent extracts analysis. The GC/MS analysis method is only suitable for the volatile compounds analysis.
Reviewer 3 Report
The autors present a paper entitled "Hepatoprotective effect of polyphenolic rich fractions from Lauridia tetragona (L.f.) R.H. Archer on acetaminophen induced liver injury". The topic is very topical and the contribution proposed by the authors is certainly interesting. The use of Lauridia tetragona and other Celastraceae as a natural remedy has been evaluated for several years, but this work evaluated the relevant content of bioactive compounds, such as the phenolic content, in relation to the hepatoprotective effect. In my opinion, this manuscript is worthy of publication on “Molecules”. In order to improve the quality of the manuscript, I suggest to make only some variations, as follows:
In abstract it’s important to better define the botanical name of the plant: Lauridia tetragona (L.f) R.H. Archer.
Briefly, more detailed results should be reported in the abstract.
Ln 58: please, add some references about Celastraceae and its relevant chemical and biological properties. Suggestions:
1) Planta Medica, (2007), 73(8), pp. 792-794 (DOI: 10.1055/s-2007-981543).
2) Planta Medica (2004), 70(7), pp. 652-656 (DOI: 10.1055/s-2004-827190).
3) Journal of Ethnopharmacology, (2009), 122(3), pp. 434-438
(DOI: 10.1016/j.jep.2009.02.003).
Ln 203-206: please, better define the suppliers of materials.
Ln 246: “The results were expressed in μM ascorbic acid per gram”, please, better define the expression of the result.
Ln 237: “results were expressed in μM ascorbic acid equivalent/ml, please, better define the expression of the result.
Reviewer 4 Report
attached
